# Score-based Generative Models with Lévy Processes

**Eunbi Yoon**[1]*, **Keehun Park**[2], **Sungwoong Kim**[3]†, **Sungbin Lim**[1, 4, 5]†
[1]Department of Statistics, Korea University
[2]Artificial Intelligence Graduate School, UNIST
[3]Department of Artificial Intelligence, Korea University
[4]LG AI Research
[5]SNU-LG AI Research Center
yon6286@gmail.com; sungbin@korea.ac.kr

## Abstract

Investigating the optimal stochastic process beyond Gaussian for noise injection in a score-based generative model remains an open question. Brownian motion is a light-tailed process with continuous paths, which leads to a slow convergence rate for the Number of Function Evaluation (NFE). Recent studies have shown that diffusion models suffer from mode-collapse issues on imbalanced data. In order to overcome the limitations of Brownian motion, we introduce a novel score-based generative model referred to as Lévy-Itô Model (LIM). This model utilizes isotropic $\alpha$-stable Lévy processes. We first derive an exact reverse-time stochastic differential equation driven by the Lévy process, then develop the corresponding fractional denoising score matching. LIM takes advantage of the heavy-tailed properties of the Lévy process. Our experimental results show LIM allows for faster and more diverse sampling while maintaining high fidelity compared to existing diffusion models across various image datasets such as CIFAR10, CelebA, and imbalanced dataset CIFAR10LT. Comparing our results to those of DDPM with 3.21 Fréchet Inception Distance (FID) and 0.6437 Recall on the CelebA dataset, we achieve 1.58 FID and 0.7006 Recall using the same architecture. LIM shows the best performance in NFE 500 with $2\times$ faster total wall-clock time than the baseline.

## 1 Introduction

Score-based diffusion models train time-dependent score models given a diffusion process and generate samples from the corresponding reverse process [39]. The feasibility of score-based diffusion models is based on the time-reversal formula of the Stochastic Differential Equations (SDEs) driven by the Brownian motion [2]. The exact form of the density function of the Brownian motion enables efficient sampling and easy theoretical handling compared to other types of noise. The score-based diffusion model is easy to train because the score function is approximated through Denoising Score-Matching (DSM). Additionally, the solution of reverse SDE can be simulated through the numerical SDE solver. This enables effective training and utilization of the model in various applications. One limitation of score-based diffusion models is that the paths produced by the SDEs need numerous iterations to achieve a data distribution, leading to slower sampling. To solve these problems, various methods have been proposed to reduce the convergence rate while maintaining to generate high fidelity samples [22]. Another effort has also been made to use distillation to reduce the number of steps [33]. In addition, it has been pointed out that Diffusion models show significant degradation in terms of fidelity and diversity when dealing with imbalanced datasets [30]. Thus, there is a need

---

*This work is done at UNIST.
†Corresponding Author.

37th Conference on Neural Information Processing Systems (NeurIPS 2023).

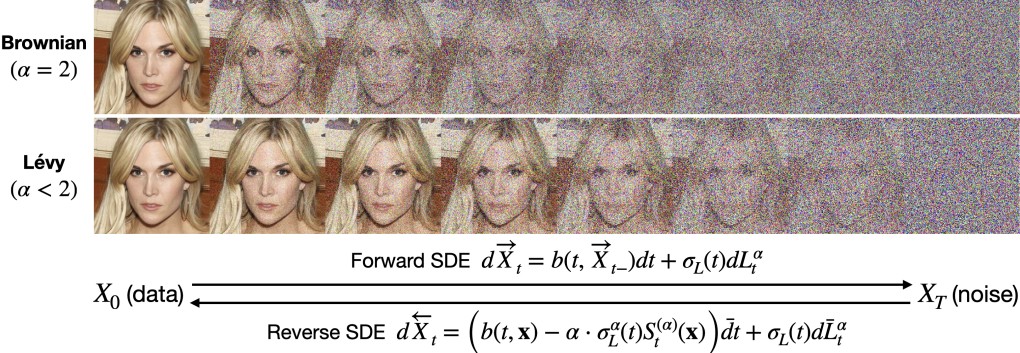

**Brownian** $(\alpha = 2)$

**Lévy** $(\alpha < 2)$

Forward SDE $d\overrightarrow{X}_t = b(t, \overrightarrow{X}_{t-})dt + \sigma_L(t)dL_t^\alpha$

$X_0$ (data)          $X_T$ (noise)

Reverse SDE $d\overleftarrow{X}_t = \left( b(t, \mathbf{x}) - \alpha \cdot \sigma_L^\alpha(t)S_t^{(\alpha)}(\mathbf{x}) \right)\bar{d}t + \sigma_L(t)d\bar{L}_t^\alpha$

Figure 1: Forward and reverse processes of Diffusion model and LIM. The formula indicates there exists a reverse-time stochastic differential equation (SDEs) corresponding to SDEs driven by isotropic $\alpha$-stable Lévy processes and can be exactly expressed by the fractional score function $S_t^{(\alpha)}(\mathbf{x})$.

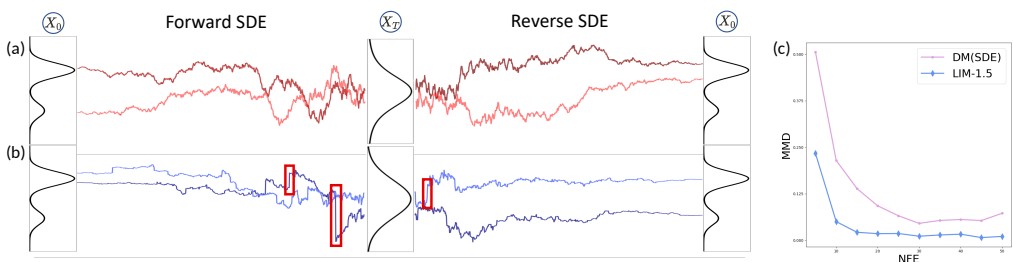

Figure 2: Trajectories of forward and reverse SDEs driven by (a) Brownian process; (b) isotropic α-stable Lévy process. Red boxes indicate large jumps generated by the heavy-tail property of $\alpha$-stable distribution. (c) Comparison of Maximum Mean Discrepancy (MMD) between the Diffusion Model (DM) [39] and the proposed model on the mixture of Gaussians.

to perform further research and develop score-based generative models that use alternative noises. Recently, there have been attempts to use new noises such as gamma distribution [24] or generalized Gaussian distribution [8]. These models have shown possibilities in expanding the range of noise processes. However, they did not propose a time-reversal formula, which is an important property for the feasibility of score-based generative models. As a result, the time-reversal formula has not yet been fully proven for alternative noises.

In this paper, we propose a novel score-based generative model, Lévy-Itô Model (LIM), which utilizes isotropic $\alpha$-stable Lévy processes as noise injection. This is the first attempt to use the isotropic $\alpha$-stable Lévy process in a score-based generative model and prove the exact time-reversal formula of SDEs driven by Lévy process. It is noted that this is also the first continuous-time model using the heavy-tailed process, which allows for the derivation of a probability fractional ODE (See more details in Appendix C). We derive a fractional score function to match the drift term of the generator of time-reversal SDEs driven by Lévy process and propose *fractional Denoising Score Matching*, which is a more general version of DSM [40] to approximate the fractional score function.

Looking at the reverse path of LIM in Figure 2, we observe large jumps occurring near the noise space. It causes the trajectory to move quickly towards the sample space, which is shown in Figure 1. As LIM has sufficient ability to move from the sample space to the noise space in spite of small noise coefficient due to large jump noises, there is a higher likelihood of reaching multiple modes more precisely. These characteristics can be observed in Figure 3, which is based on the simplest form of an imbalanced dataset. LIM's superiority can be observed not only from the ratio aspect but also from FID and MMD, which are calculated directly without using any embeddings for the sets of true samples and generated samples. LIM shows that the distribution $p_\theta$ of its generated samples is similar to the ground truth distribution $p_{\text{data}}$.

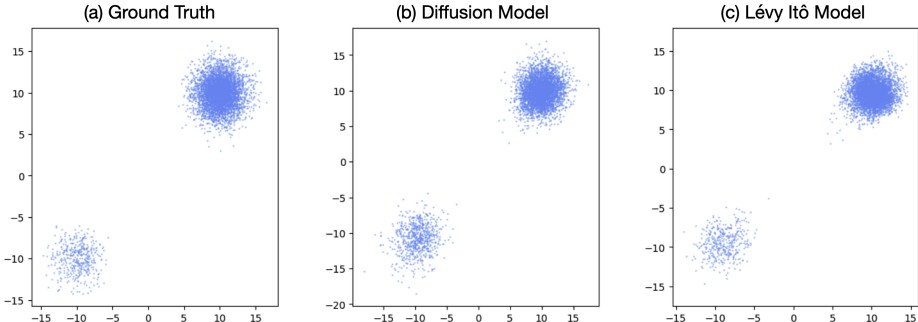

Figure 3: Mode estimation for the synthetic data. The dataset consists of two mixtures of Gaussian distributions, with a ground truth ratio of 10:1. The blue clusters in subplots (a) is the ground truth and each (b) and (c) represents the distribution of generated samples in 2D by the diffusion model and LIM. In subplot (b), it can be seen that for the Diffusion model, the generated samples have a ratio of 5.6:1, which is quite different from the ratio of ground truth in subplot (a). However, in subplot (c), for LIM, the generated samples have a ratio of 11.1:1, which is relatively close to the ground truth. This similarity can be observed not only from the ratio aspect but also from FID (Fréchet Inception Distance) and Maximum Mean Discrepancy (MMD). The diffusion model shows FID of 8.312 ± 0.904 and MMD of 0.02 ± 0.002. On the other hand, the LIM achieves FID of **0.663 ± 0.376** and MMD of **0.026 ± 0.003**.

We draw inspiration from the observation that the time-reversal formula for SDEs is known not only for diffusion processes but also for SDEs driven by Lévy processes containing jumps [7]. In addition, the invariant measure of the Euler scheme has been shown to converge to the invariant measure of the solution of SDEs driven by Lévy process [6]. Moreover, there is an example of using Lévy processes in Markov Chain Monte Carlo (MCMC), where the corresponding research demonstrated a faster convergence rate for NFE both theoretically and experimentally [37]. However, it cannot be directly extended to SDEs driven by multivariate stable processes and use an approximation to calculate the score function of a data distribution, which degrades the potential performance. These observations motivate us to consider using Lévy processes in score-based generative models.

Our experimental results show LIM allows for faster and more diverse sampling while maintaining high fidelity compared to diffusion models across various image datasets such as CIFAR10, CelebA, and imbalanced dataset CIFAR10LT. We realize that the heavy-tail property of the Lévy process allows us to overcome the limitations of Brownian motion with respect to mode estimation, sample diversity, and convergence rate for NFE. Through the above experiments, we emphasize we have further expanded the scope of score-based generative models.

The contributions of our paper can be summarized as follow:

- We propose a novel score-based generative model, Lévy-Itô Model (LIM), which utilizes isotropic $\alpha$-stable Lévy processes as noise injection.

- We derive an exact reverse-time stochastic differential equation driven by the Lévy process and develop the corresponding fractional Denoising Score Matching.

- We validate that LIM allows for faster and more diverse sampling while maintaining high fidelity compared to existing diffusion models across various image datasets and shows more robustness against imbalanced data.

## 2   Related works

Diffusion models have shown improvements in performance through various approaches such as incorporating guidance [18], introducing new architectures [29], and proposing novel training methods [13]. Additionally, various attempts to reduce convergence rates have been proposed, such as using ODE solvers [23], or Fourier Neural Operator [42]. Despite these advancements, diffusion models still face inherent limitations such as slow convergence rates and the mode-collapse issue in imbalanced datasets [30]. There have been attempts to use new noises in score-based generative models. The

denoising generative model with gamma distribution is one such example [24], which shows faster convergence for NFE than score-based diffusion models. [8] proposes using heavy-tailed noises in score-based generative models by extending the noise injection to generalized Gaussian distributions. The authors state that the heavy-tailed DSM is more robust against imbalance tasks than score-based diffusion models. They commonly use a variant of the Denoising Diffusion Probabilistic Model (DDPM) structure to apply alternative noises. These models have shown the potential to expand the range of noise processes, however, their performance in terms of the Fréchet Inception Distance (FID) is not as high as that of score-based diffusion models.

Since these distributions follow a Lévy process [11, 41], the drift term for the time-reversal formula can be computed according to [7]. For isotropic $\alpha$-stable Lévy processes, which follow symmetric and stable distributions, an exact drift term can be computed from the time-reversal formula. Training can be stable and accurate without the needs for integration. Furthermore, with the knowledge of the exact drift term, a reverse sampling formula ensuring convergence can be derived. In contrast, the drift terms used in Heavy-tailed DSM and Denoising Diffusion Gamma Models appear as an integral, making it difficult to derive an exact form. Even if integration is used during training or approximations are applied to the drift term, accumulating errors lead to inaccurate and unstable training. Moreover, when using an approximation for the drift term to make training practical, there's no assurance that the distance between the distribution of actual data, $p_{\text{data}}$ and the distribution obtained from reverse sampling, $p_\theta$ aligns within a certain range. For Heavy-tailed DSM, despite proposing a modified score function corresponding to the Generalized Gaussian distribution and providing a training method, there is no theoretical foundation proving that the modified score function converges to $p_{\text{data}}$ through reverse sampling. Similarly, Denoising Diffusion Gamma Models also fail to propose an exact drift term and present limitations by not providing theoretical evidence for the convergence of reverse sampling using the proposed score function. As a result, the FID performances of these two papers are significantly lower than that of DDPM [15].

## 3 Background

### 3.1 Isotropic alpha-stable distributions

Let $\alpha \in (0, 2]$ be a characteristic exponent that determines the decay rate, and $\gamma$ be a scale parameter. For $d \in \mathbb{N}$, $\mathcal{S}\alpha\mathcal{S}^d(\gamma)$ denotes the $d$-dimensional **isotropic $\alpha$-stable distribution**. If a random variable $X \in \mathbb{R}^d$ follows $\mathcal{S}\alpha\mathcal{S}^d(\gamma)$, then the characteristic function of $X$ is $\mathbb{E}[e^{i\langle \mathbf{u}, X \rangle}] = e^{-\gamma^\alpha ||\mathbf{u}||^\alpha}$ and has heavy-tail properties with tail index $\alpha$, i.e., $\mathbb{P}(||X|| > r) \sim r^{-\alpha}$. A closed-form of density

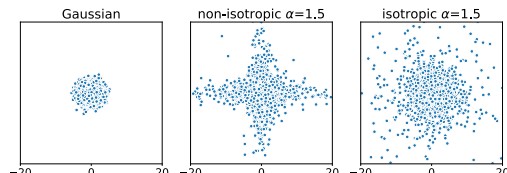

Figure 4: Comparison of samples from Gaussian, non-isotropic, and isotropic $\alpha$-stable distribution.

function for $\mathcal{S}\alpha\mathcal{S}^d(\gamma)$ is unknown, except for $\alpha = 1$, $\alpha = 1.5$ and $\alpha = 2$ cases[3]. Note that the behavior of isotropic $\alpha$-stable distribution is different from *non-isotropic* $\alpha$-stable in multi-dimension since the components of isotropic $\alpha$-stable are not independent unless $\alpha = 2$. Contrary to [43] which can not be extended to SDEs that are driven by $\alpha$-stable Lévy processes, we utilize isotropic $\alpha$-stable processes for driving noise in SDEs. See Figure 4 for the comparison between sample distributions of Gaussian, non-isotropic $\alpha$-stable, and isotropic $\alpha$-stable random variable.

### 3.2 Lévy processes

An $\mathbb{R}^d$-valued stochastic process $(L_t)_{t \geq 0}$ with $L_0 = 0$ is called Lévy process if (i) $L_t$ has independent increments, (ii) $L_t$ has stationary increments, and (iii) $L_t$ has stochastically continuous sample paths. Thus, Lévy process may have at most countably many discontinuous points, also known as jumps [3]. If for all $s < t$, $(L_t - L_s) \stackrel{d}{=} L_{t-s}$ follows $\mathcal{S}\alpha\mathcal{S}^d((t-s)^{1/\alpha})$, where $\stackrel{d}{=}$ means that the two processes have the same law, then $L_t$ is referred to as a $d$-dimensional isotropic $\alpha$-stable Lévy process, denoted by $L_t^\alpha$. Notably, $L_t^\alpha$ is a prototypical pure jump process. The $\alpha$-stable distribution has a heavy-tailed property, meaning that the frequency of large jumps increases as $\alpha$ decreases. According to the

---

[3]When $\alpha = 2$, it holds $\mathcal{S}\alpha\mathcal{S}^d(\gamma) = \mathcal{N}^d(0, \sqrt{2}\gamma\mathbf{I})$ where $\mathcal{N}$ is a Gaussian distribution and $I \in \mathbb{R}^{d \times d}$ is the identity operator

Lévy-Itô decomposition [21], we can split the sample paths of Lévy process into drift parts, Brownian process parts, and jump parts. The focus of this paper is on rotationally invariant pure-jump Lévy processes, whose Lévy measure is $\nu(\mathrm{d}\mathbf{x}) \sim ||\mathbf{x}||^{-d-\alpha}\mathrm{d}\mathbf{x}$.

## 4 Lévy-Itô Model

### 4.1 Time-Reversal of SDEs driven by Lévy processes

In this section, we provide a time-reversal theory for SDEs driven by Lévy processes as follows:

$$d\overrightarrow{X}_t = b(t, \overrightarrow{X}_{t-})dt + \sigma_L(t)dL_t^\alpha, \quad t \in [0, T] \tag{1}$$

where $\overrightarrow{X}_{t-}$ denotes the left limit of $\overrightarrow{X}$ at time $t$ which implies the *càdlàg* (right conti. with left limits) solution. We define *fractional score function* $S_t^{(\alpha)}$ by,

$$S_t^{(\alpha)}(\mathbf{x}) := \frac{\Delta^{\frac{\alpha-2}{2}}\nabla p_t(\mathbf{x})}{p_t(\mathbf{x})}, \quad \mathbf{x} \in \mathbb{R}^d, t \in [0, 1] \tag{2}$$

where $\Delta^{\frac{\gamma}{2}}$ denote the fractional Laplacian of order $\frac{\gamma}{2}$ for $\gamma \in (-1, 2)$ [27]. Precisely, $\Delta^{\frac{\gamma}{2}}f(\mathbf{x}) := \mathcal{F}^{-1}\{||\mathbf{u}||^\gamma \mathcal{F}\{f\}(\mathbf{u})\}$ where $\mathcal{F}$ and $\mathcal{F}^{-1}$ are the Fourier and the inverse Fourier transform, respectively. The *time-reversal* of $\overrightarrow{X}_t$ is defined by $\overleftarrow{X}_t := \overrightarrow{X}_{t-} - \overrightarrow{X}_{T-}$ where $t$ runs backward from $T$ to $0$. Note that $\overleftarrow{X}_t$ is a Lévy process with respect to natural filtration when time flows forwards [17], which is the *càglàd* (left conti. with right limits). Our first main result is the derivation of the *reverse-time SDE* corresponding to (1).

**Theorem 4.1** (Time-Reversal of SDE). *Let $1 < \alpha \leq 2$. Suppose $b(t, \mathbf{x})$ is locally bounded and the marginal distribution of $(X_t)_{t \in [0,T]}$ has a density function $p_t(\mathbf{x})$.*

$$d\overleftarrow{X}_t = \left(b(t, \overleftarrow{X}_{t+}) - \alpha \cdot \sigma_L^\alpha(t)S_t^{(\alpha)}(\overleftarrow{X}_{t+})\right)\bar{d}t + \sigma_L(t)d\bar{L}_t^\alpha + d\bar{Z}_t, \tag{3}$$

*where $\bar{d}t$ is an infinitesimal negative timestep, $\bar{L}_t^\alpha$ is the backward version of the isotropic $\alpha$-stable Lévy process, and $\bar{Z}_t$ is the backward version of a Lévy-type stochastic integral $Z_t$ such that $\mathbb{E}[Z_t] = 0$ with finite variation.*

See more detail in Theorem B.1 for the definition of $Z_t$. Now we propose a score-based generative model with Lévy process $L_t^\alpha$, refers to **Lévy-Itō Model** (LIM). In [39], the VP-SDE formula is defined as $d\overrightarrow{X}_t = -\frac{\beta(t)}{2}\overrightarrow{X}_t dt + (\beta(t))^{\frac{1}{2}}dB_t$ and $\beta(t)$ is set to a linear function $\beta(t) = (\beta_1 - \beta_0)t + \beta_0$. On the other hand, we extended [39]'s formula to the SDE driven by isotropic $\alpha$-stable Lévy process, $d\overrightarrow{X}_t = -\frac{\beta(t)}{\alpha}\overrightarrow{X}_t dt + (\beta(t))^{\frac{1}{\alpha}}dL_t^\alpha$, where we simulate the noise term in (3) using only $L_t^\alpha$. This choice is motivated by the fact that, in contrast to $\sigma_L(t)dL_t^\alpha$, which exhibits infinite variation, $dZ_t$ has a mean of 0 and finite variation, rendering it comparatively negligible. The specific $\beta(t)$ in our paper is derived from the cosine schedule proposed in [25] and tailored to LIM as $\beta(t) = -\alpha \frac{d\log(\cos(\frac{\pi t}{2}))}{dt}$ for setting a forward SDE with $b(t, \mathbf{x}) = -\frac{\beta(t)}{\alpha}\mathbf{x}$ and $\sigma_L(t) = \beta^{\frac{1}{\alpha}}(t)$ in (1). Then the solution $\overrightarrow{X}_t$ has the same law with $a(t)\overrightarrow{X}_0 + \gamma(t)\boldsymbol{\epsilon}$ where $\boldsymbol{\epsilon} \sim \mathcal{S}\alpha\mathcal{S}^d(1)$, $a(t) = \exp(-\int_0^t \frac{\beta(s)}{\alpha}ds)$, and $\gamma(t) = (1 - a^\alpha(t))^{\frac{1}{\alpha}}$. As $b(t, \mathbf{x}) = -\frac{\beta(t)}{\alpha}\mathbf{x}$ satisfies the condition of Theorem 4.1, we can obtain the reverse-time SDE of the beta-schedule version,

$$d\overleftarrow{X}_t = \left(-\frac{\beta(t)}{\alpha}\overleftarrow{X}_{t+} - \alpha \cdot \beta(t) \cdot S_t^{(\alpha)}(\overleftarrow{X}_{t+})\right)\bar{d}t + (\beta(t))^{\frac{1}{\alpha}}d\bar{L}_t^\alpha. \tag{4}$$

For $t < s$, the solution of equation (4) can be represented as an integral, utilizing the semi-linear structure of the reverse SDE,

$$\mathbf{x}_t = \frac{a(t)}{a(s)}\mathbf{x}_s - \alpha^2 \cdot a(t)\int_s^t \frac{d}{du}\left(\frac{1}{a(u)}\right)S_u^{(\alpha)}(\mathbf{x}_u)du + \int_s^t (\beta(u))^{\frac{1}{\alpha}}\frac{a(t)}{a(u)}d\bar{L}_t^\alpha. \tag{5}$$

We can get faster sampling method by using an approximation for the second term $\int_s^t \frac{\beta(u)}{a(u)} S_u^{(\alpha)}(\mathbf{x}_u) du$. This term can be approximated as $\left( \int_s^t \frac{\beta(u)}{a(u)} du \right) \cdot S_s^{(\alpha)}(\mathbf{x}_s)$ and it is possible to calculate $\int_s^t \beta(u) \frac{a(t)}{a(u)}$, the scale parameter $\gamma$ of which follows $\gamma^\alpha = \left| \int_s^t \frac{d}{du} \left( e^{-\int_u^t \beta(k)} \right) \right|$.

**Corollary 4.1** (Variant of Euler-Maruyama with dynamic time increment). *Suppose the fractional score function in the SDE given by equation ([4](#)) satisfies the conditions stated in Theorem [E.2](#), and $a(t), \gamma(t)$ are bounded. Then, there exists a Markov chain $(\mathbf{x}_t)$ that follows:*

$$\mathbf{x}_t = \frac{a(t)}{a(s)} \mathbf{x}_s + \alpha^2 \left( \frac{a(t)}{a(s)} - 1 \right) S_s^{(\alpha)}(\mathbf{x}_s) + \left( \left( \frac{a(t)}{a(s)} \right)^\alpha - 1 \right)^{\frac{1}{\alpha}} \boldsymbol{\epsilon}. \tag{6}$$

*Here, $\boldsymbol{\epsilon} \sim \mathcal{S}\alpha\mathcal{S}^d(1)$ for $s > t$, and $\Delta t = s - t \ll 1$. As a result of the conditions being satisfied, the Wasserstein-1 distance between the invariant measures of the solution of Equation ([4](#)) and $(\mathbf{x}_t)$ is bounded by $(\Delta t)^{\frac{1}{\alpha}}$ [6]. (See more detail in Appendix [E](#))*

The paper [6] outlines a way to find bounds on the Wasserstein-1 distance by determining bounds for the approximation of the drift term, $J_1$, and the approximation of the stochastic term, $J_2$. In the equation, only the bound for the approximation of the drift term, $J_1$, is utilized. Furthermore, since the law of a weak solution is same to the law of the strong solution, the Wasserstein-1 distance between the invariant measures for the strong solution and the approximation can be used for the that of the weak solution and the approximation as well. In addition, we can also find a probability ODE corresponding to ([1](#)).

**Theorem 4.2** (Probability Fractional ODE). *Let $b(t, \mathbf{x}) : \mathbb{R}^d \to \mathbb{R}$ and $\sigma_L(t) : \mathbb{R} \to \mathbb{R}$ be functions that satisfy the Lipschitz condition as stated in [35]. For the SDE following ([1](#)), the solution $(X_t)_{t \in [0,T]}$ of the SDE satisfies the following ODE:*

$$d\vec{X}_t \stackrel{d}{=} \left( b(t, \vec{X}_{t-}) - \sigma_L^\alpha(t) S_t^{(\alpha)}(\vec{X}_{t-}) \right) dt. \tag{7}$$

Due to its semilinear structure, the solution to ([7](#)) can be represented as an integral. This is shown in Lemma [D.2](#). The Euler method can then be used to find the solution.

**Corollary 4.2** (Deterministic ODE sampling). *If the drift term in ([7](#)) is Lipschitz continuous and the solution $\mathbf{x}_t$ has a bounded second derivative, then a sequence $(\mathbf{x}_t)$ can be obtained using the Euler-scheme:*

$$\mathbf{x}_t = \frac{a(t)}{a(s)} \mathbf{x}_s + \alpha \cdot \left( \frac{a(t)}{a(s)} - 1 \right) \cdot S_s^{(\alpha)}(\mathbf{x}_s) \tag{8}$$

*where $s > t$. When the step size is $\Delta t$, the global truncation error is bounded by $O(\Delta t)$ [5].*

### 4.2 Estimating Fractional Score Function

Let $q_\alpha(\mathbf{x})$ be the density of $\mathcal{S}\alpha\mathcal{S}^d(1)$. A weak solution of the beta-schedule version is $\vec{X}_t = a(t)\vec{X}_0 + \gamma(t)\boldsymbol{\epsilon}$ where $\boldsymbol{\epsilon} \sim \mathcal{S}\alpha\mathcal{S}^d(1)$ with the following transition density:

$$p_t(\mathbf{x}_t | \mathbf{x}_0) = \frac{1}{\gamma^d(t)} q_\alpha \left( \frac{\mathbf{x}_t - a(t)\mathbf{x}_0}{\gamma(t)} \right). \tag{9}$$

We train the *fractional score model* $\mathbf{s}_t(\mathbf{x}_t; \theta)$ which estimates $S_t^{(\alpha)}(\mathbf{x}_t)$ at the optimal point. The method of approximating the fractional score function $S_t^{(\alpha)}(\mathbf{x}_t)$ can be equivalent to estimating the fractional score function $S_t^{(\alpha)}(\mathbf{x}_t | \mathbf{x}_0)$, which replaces the fractional score function of term $p_t(\mathbf{x}_t)$ in ([2](#)) with that of the transition density ([9](#)).

**Theorem 4.3** (Fractional Denoising Score Matching). *For a parameter $\theta$, two loss functions $L_1(\theta)$ and $L_2(\theta)$ are given as follows:*

$$L_1(\theta) = \mathbb{E}_{t, \mathbf{x}_t \sim p_t(\mathbf{x}_t)} \| \mathbf{s}_t(\mathbf{x}_t; \theta) - S_t^{(\alpha)}(\mathbf{x}_t) \|_2^2. \tag{10}$$

$$L_2(\theta) = \mathbb{E}_{t, (\mathbf{x}_0, \mathbf{x}_t) \sim p_t(\mathbf{x}_0, \mathbf{x}_t)} \| \mathbf{s}_t(\mathbf{x}_t; \theta) - S_t^{(\alpha)}(\mathbf{x}_t | \mathbf{x}_0) \|_2^2. \tag{11}$$

*Then, there exists a constant $C$ satisfying $L_1(\theta) = L_2(\theta) + C$ so that the optimization of two loss functions with respect to $\theta$ is equivalent.*

The drift term of the reverse-time SDEs can be accurately determined by training $\mathbf{s}_t(\mathbf{x}_t; \theta)$ to match $S_t^{(\alpha)}(\mathbf{x}_t|\mathbf{x}_0)$, as stated in Theorem 4.3. Moreover, if we have the jump-type Lévy process $L_t^\alpha$, we can calculate $S_t^{(\alpha)}(\mathbf{x}_t|\mathbf{x}_0)$ explicitly as a linear expression.

**Theorem 4.4.** *Let $1 < \alpha < 2$. If the transition density function $p_t(\mathbf{x}_t|\mathbf{x}_0)$ follows (9), then the fractional score function of $p_t(\mathbf{x}_t|\mathbf{x}_0)$ can be represented as a linear form such that*

$$S_t^{(\alpha)}(\mathbf{x}_t|\mathbf{x}_0) = \frac{\Delta^{\frac{\alpha-2}{2}}\nabla p_t(\mathbf{x}_t|\mathbf{x}_0)}{p_t(\mathbf{x}_t|\mathbf{x}_0)} = -\frac{1}{\gamma^{\alpha-1}(t)} \cdot \frac{1}{\alpha} \cdot \left(\frac{\mathbf{x}_t - a(t)\mathbf{x}_0}{\gamma(t)}\right). \tag{12}$$

Theorem 4.4 implies that the computational cost of training a fractional score model $\mathbf{s}_t(\mathbf{x}_t; \theta)$ is similar to DSM [40].

### 4.3 Loss function

We set beta schedule as cosine beta schedule $\beta(t)$ [25] , following $\beta(t) = \frac{\alpha}{1+s}\frac{\pi}{2}\tan\left(\frac{t+s}{1+s}\frac{\pi}{2}\right)$ for small number $0 < s \ll 1$. Then, the drift term $a(t) = \frac{\cos(\frac{t+s}{1+s}\frac{\pi}{2})}{\cos(\frac{s}{1+s}\frac{\pi}{2})}$ and the diffusion term $\gamma(t) = (1 - a(t)^\alpha)^{\frac{1}{\alpha}}$. We use the U-net architecture as in DDPM [15] for training and apply the loss (11) to train $\mathbf{s}_t(\mathbf{x}_t; \theta)$. For $\epsilon \sim \mathcal{S}\alpha\mathcal{S}^d(1)$ and $x_0 \sim p_{\text{data}}$, $\mathbf{x}_t = a(t)\mathbf{x}_0 + \gamma(t)\epsilon$, $U(0, 1)$ denotes a uniform distribution. Then, the loss with the relative weight $\gamma(t)$ is defined as

$$L(\theta; \gamma(t)) = \mathbb{E}_{t \sim U(0,1)}\mathbb{E}_{x_0 \sim p_{\text{data}}}\mathbb{E}_{\epsilon \sim \mathcal{S}\alpha\mathcal{S}^d(1)}\left\|\gamma^{\alpha-1}(t)\mathbf{s}_t(\mathbf{x}_t; \theta) + \frac{\epsilon}{\alpha}\right\|_2^2. \tag{13}$$

The fractional score model $\mathbf{s}_t(\mathbf{x}_t; \theta)$ learns the law of the solution $X_t$ for each $t$ because the beta-schedule version solution used during training is the weak solution of the given forward SDEs.

## 5 Experiments

We employ continuous time step throughout our experiments and set NFE to a fixed value of 500 in all cases. Given that Lévy processes exhibit a higher occurrence of extreme values compared to Gaussian noise, we opt for the smooth $L_1$ Loss instead of the $L_2$ Loss to ensure stable training. We use the modified VP-SDE to fit the Lévy process and quadratic timestep during reverse sampling.

### 5.1 Mode Estimation

We conduct the class imbalance experiment for mode estimation with two kinds of datasets. First is customized imbalanced CIFAR10 with a ratio given of 1:2:3:4:5. We choose t-distributed stochastic neighbor embedding (t-SNE), which reduces complex data of high dimensions to two dimensions. In table 1 and Fiure 5, the mode estimation of LIM is much closer to training data than the diffusion model. Also, the corresponding FID, Recall, and MMD of LIM show improved performances. The second dataset is CIFAR10LT including 10 classes, which follow heavy-tailed distribution [4]. Table 3 shows that as we move from class 0 with a high ratio to class 9 with a low ratio, the overall performance tends to decrease. However, LIM shows better performances compared to the diffusion model with respect to FID and Recall. These results demonstrate that LIM is more robust for mode estimation in imbalanced datasets, and alleviates the mode-collapse issue.

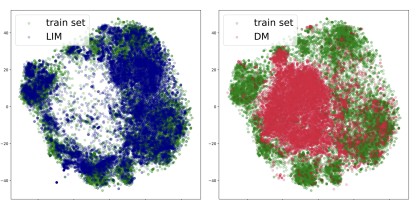

Figure 5: t-SNE and evaluation in the imbalanced CIFAR10. LIM (blue), DM (red), and training data (green).

### 5.1.1 Sample Diversity

---

| Metric | Model | airplane | automobile | bird | cat | deer | dog | frog | horse | ship | truck |
|---|---|---|---|---|---|---|---|---|---|---|---|
| FID↓ | DM | 13.92 | 8.87 | **14.58** | **15.44** | 10.35 | **16.48** | **13.19** | 11.05 | 9.50 | 8.38 |
| | LIM | 13.92 | **8.24** | 15.02 | 16.07 | **10.05** | 18.24 | 13.23 | **9.77** | **9.18** | **7.76** |
| Recall↑ | DM | 0.6646 | 0.6238 | 0.6968 | 0.6076 | 0.6936 | 0.6792 | 0.6816 | 0.6734 | 0.6342 | 0.6098 |
| | LIM | **0.6912** | **0.6516** | **0.7138** | **0.6272** | **0.6976** | **0.7212** | **0.7082** | **0.7040** | **0.6472** | **0.6366** |

Table 2: Results of conditional generation on **CIFAR10**($32 \times 32$).

| Metric | Model | airplane | automobile | bird | cat | deer | dog | frog | horse | ship | truck |
|---|---|---|---|---|---|---|---|---|---|---|---|
| FID↓ | DM | 22.00 | 12.69 | 30.55 | 37.14 | 22.08 | 49.31 | 38.67 | 44.82 | 50.97 | 36.08 |
| | LIM | **15.594** | **9.5131** | **19.928** | **25.66** | **15.90** | **33.5393** | **29.46** | **24.88** | **30.34** | **34.302** |
| Recall↑ | DM | 0.6412 | 0.5716 | 0.6500 | 0.5200 | 0.6240 | 0.5828 | **0.5796** | **0.6246** | 0.5890 | 0.6046 |
| | LIM | **0.6768** | **0.6118** | **0.6778** | **0.5696** | **0.6248** | **0.6578** | 0.5686 | 0.6100 | **0.5906** | **0.6084** |

Table 3: Results of conditional generation on **CIFAR10LT**($32 \times 32$).

We choose Recall as an evaluation metric for sample diversity. In Table 5, we can see that LIM exceeds other diffusion models in Recall. Moreover, even in the conditional generation, it shows higher Recall for each class, proving that the generated samples are more diverse than those of the diffusion model (Table 2).

| Metric | LIM | DM |
|---|---|---|
| FID↓ | **21.07** | 62.62 |
| Recall↑ | **0.5549** | 0.5002 |
| MMD↓ | **0.00416** | 0.01396 |

Table 1: Imbalanced CIFAR10 results.

**Imputation** The imputation refers to the task of restoring the masked regions in an image. To measure the diversity of samples, we use Learned Perceptual Image Patch Similarity (LPIPS) as a metric, where a higher LPIPS indicates greater diversity and a lower value implies more similarity between samples. For each original image $x$, we generate 10 imputed images $\{\hat{x}_i\}_{i=1}^{10}$ and calculate the average of LPIPS for the 10 samples. As shown in Figure 6, the diffusion model tends to generate similar eyes and mouths in the masked region, while LIM produces a much wider range of diverse images and shows higher LPIPS.

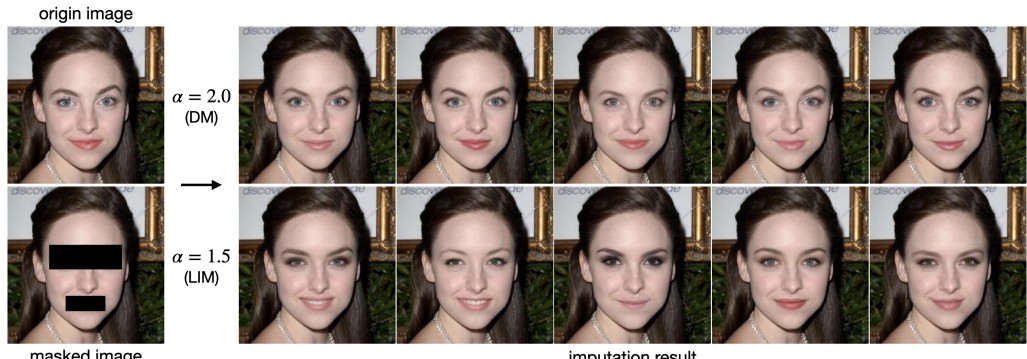

Figure 6: Imputaion images on CelebA-HQ dataset. Diffusion model(DM) generates only similar eyes and lips, while LIM generates a much more diverse range of shape and color of eyes and lips. LPIPS↑ of DM is 0.0132, but LIM is **0.0197**.

## 5.2 Convergence rate for NFE

We compare LIM and diffusion models on CIFAR10 ($32 \times 32$) and CelebA ($64 \times 64$) dataset in terms of convergence rate for NFE. Figure 7 shows that FID score of LIM drops at earlier NFE than the baselines in SDE-based sampling. Moreover, it shows a competitive convergence rate for NFE compared to ODE-based models. These experimental results demonstrate that LIM is capable of converging to the sample space faster than the baselines. See Appendix G.6 to figure out the wall clock time per NFE on various datasets.

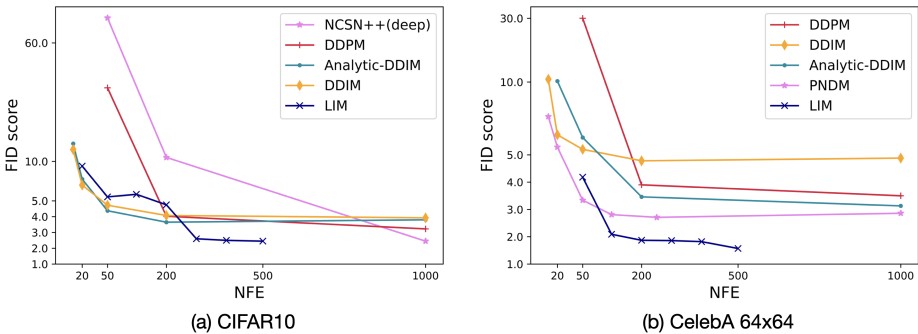

Figure 7: Convergence rate for NFE measured by FID score on CIFAR10 and CelebA.

## 5.3 Sample Quality

Table 5 displays the performance of LIM in comparison to previous approaches for unconditional image generation on CIFAR10 ($32 \times 32$) and CelebA ($64 \times 64$). We measure FID scores and Recall using reverse sampling and generate 50k samples for all datasets. For CelebA, we randomly select 50k samples from the training dataset with 5 iterations. We evaluate NLL using a uniformly dequantized test dataset and calculate the log-likelihood through the average of 5 iterations using the ODE solver [39]. Since it is difficult to numerically calculate the density function of the isotropic $\alpha$-stable distribution in higher dimensions ($d > 100$) [26], we approximate the density function with non-isotropic $\alpha$-stable distributions to evaluate the NLL of LIM. For CIFAR10, we train 2 backbone models: DDPM [15] and NCSN++(deep) [39]. See Appendix G for more detail.

The performance of LIM can differ with different $\alpha$. Indeed, this seems intuitively natural since images with low resolution less prefer large jumps, while images with high resolution and multiple modalities can benefit exploration from large jumps. Table 4 provides a comparison of LIM's performance across different $\alpha$ values. Although the overall tendency of declining FID with lower values remains consistent, it appears that differences in FID it-

| $\alpha$ | CIFAR10 | CelebA |
|---|---|---|
| 1.2 | 5.15 | 2.99 |
| 1.5 | 2.86 | 1.57 |
| 1.8 | 2.44 | 3.21 |

Table 4: $\alpha$ selection on CIFAR10 ($32 \times 32$) and CelebA ($64 \times 64$).

self arises when using different architectures on CIFAR10 ($32 \times 32$). In spite of using half of NFE used in [39], LIM shows competitive FID score, Recall, and NLL. Especially in CelebA ($64 \times 64$), LIM achieves FID score of 1.57 and Recall of 0.7007. It is an impressive performance since we do not use any other guidance or improved model architecture. We also conducted performance comparisons between diffusion model [39] and LIM using the ADM architecture [9] on ImageNet ($64 \times 64$). Diffusion model [39] achieves a FID of 14.23, Precision of 0.6711, and Recall 0.6932. In contrast, LIM achieves a FID of 12.97, Precision of 0.6782, and Recall of 0.6782. For comparing the performance on a high-resolution dataset, we chose the DDPM architecture [15], and trained LIM and diffusion model [39] on the CelebA-HQ dataset ($256 \times 256$). This indicates that when it comes to CelebA-HQ, LIM attains an FID score of 7.76 with 500 NFE, whereas the diffusion model achieves a higher FID of 11.87 with 1000 NFE based on DDPM architecture [15].

## 5.4 Deterministic ODE sampling

We have the option to utilize DDIM inference for our model. When we use a pretrained model on CelebA (64x64), Table 6 displays the FID scores acquired by DDIM and LIM-DDIM for various NFEs. The results demonstrate that LIM-DDIM outperforms DDIM in terms of performance.

| NFE | 20 | 50 | 100 | 200 |
|---|---|---|---|---|
| DDIM | 6.64 | 5.23 | - | 4.78 |
| LIM-DDIM | 6.73 | 4.80 | **3.95** | - |

Table 6: DDIM and LIM-DDIM with $\alpha = 1.5$ on CelebA ($64 \times 64$).

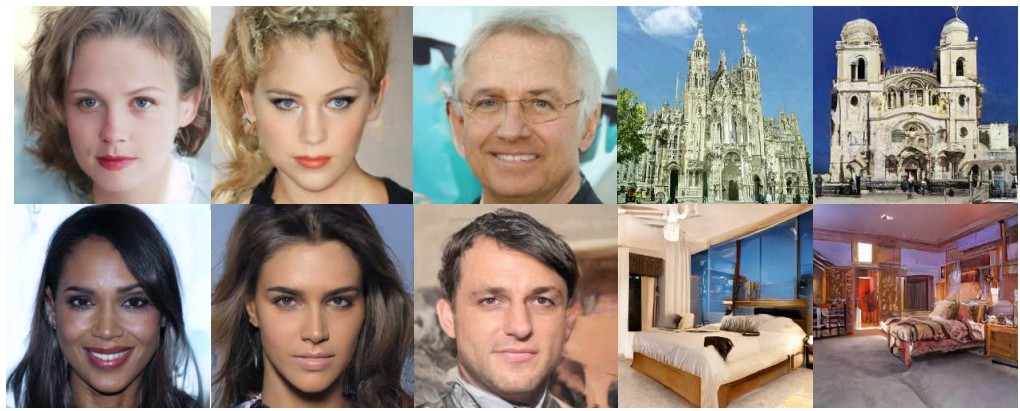

Figure 8: Generated samples on CelebA-HQ $256 \times 256$, LSUN-Church $256 \times 256$, and LSUN-Bedroom $256 \times 256$.

| Model | Noise | NFE | FID↓ | Recall↑ | †NLL(bits/dim)↓ |
|---|---|---|---|---|---|
| **CIFAR10** $(32 \times 32)$ | | | | | |
| DDPM [15] | Gaussian | 1000 | 3.17 | - | $\leq 3.70$ |
| DDPM cont. (VP) [39] | Gaussian | 1000 | 3.24 | 0.6784 | 3.21 |
| NCSN++ cont. (deep, VP) [39] | Gaussian | 1000 | **2.44** | 0.6860 | 3.13 |
| LIM-DDPM cont. (Ours) | Lévy ($\alpha$=1.8) | 500 | 3.37 | 0.6874 | 3.08 |
| LIM-NCSN++ cont. (deep, Ours) | Lévy ($\alpha$=1.8) | 500 | **2.44** | **0.6960** | **2.97** |
| **CelebA** $(64 \times 64)$ | | | | | |
| DDPM [15] | Gaussian | 1000 | 3.50 | - | - |
| DDPM cont. (VP) [39] | Gaussian | 1000 | 3.21 | 0.6437 | 2.39 |
| LIM-DDPM cont. (Ours) | Lévy ($\alpha$=1.5) | 500 | **1.57** | **0.7007** | **2.23** |

Table 5: Unconditional generation on CIFAR10 $(32 \times 32)$ and CelebA $(64 \times 64)$. †NLL is approximated by using non-isotropic $\alpha$-stable distribution as a prior density.

## 6   Limitation

Many diffusion models have been conducted to find the best model architectures such as NCSN++ [39] and ADM [9]. In contrast, Lévy process differs significantly from Brownian motion. Therefore, it is uncertain whether using the same model architecture that has been used previously is the best approach for LIM. Although, in this study, we use the same structure of the existing diffusion model to compare the pure effect of the Lévy process, there is a need to find a model architecture that is suitable for the Lévy process in order to improve performance further.

## 7   Conclusion

The aim of this study is to expand the variety of noise distributions utilized in score-based generative models by introducing an exact time-reversal formula for SDEs with Lévy processes. We propose a new score-based generative model, Lévy-Itô Model (LIM). The effectiveness of this model has been verified by empirical results, which show that LIM performs well with various image dataset. As a result, this study provides a solution and demonstrates the potential for utilizing a wider range of non-Gaussian Markov processes in score-based generative models.

## Acknowledgments and Disclosure of Funding

This work was supported by LG AI Research. This work was also supported by Institute of Information & communications Technology Planning & Evaluation(IITP) grant funded by the Korean government(MSIT)(No. 2022-0-00612, Geometric and Physical Commonsense Reasoning based Behavior Intelligence for Embodied AI; No. 2019-0-00079, Artificial Intelligence Graduate School Program, Korea University), National Research Foundation of Korean(NRF) funded by the Korean government(MSIT)(2021R1C1C1009256), and a grant from Korea University (K2304791). This work also was supported by Artificial intelligence industrial convergence cluster development project funded by the Ministry of Science and ICT(MSIT, Korea) and Gwangju Metropolitan City. The authors especially thank Prof. Nicolas Privault for his interest in our paper and valuable discussions that helped us correct errors in our main result. The first author would like to thank Woo-Joo Kim for insightful discussion and anonymous reviewers for their constructive discussion and editorial efforts.

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
