# OpenReview forum: "Score-based Generative Models with Lévy Processes"
_NeurIPS.cc/2023/Conference — NeurIPS 2023 spotlight_

### Official Review · Reviewer_Z4Ey · 2023-07-03

**Soundness:** 4 excellent
**Presentation:** 4 excellent
**Contribution:** 3 good
**Rating:** 7
**Confidence:** 4

**Summary:**

Score-based generative models (SBGMs) generally employ Brownian motion, also known as the Wiener process, for noise injection. However, using Brownian motion in SBGMs often leads to issues such as mode collapse or slow sampling. To address these problems, the authors propose SBGMs with an isotropic α-stable Levy distribution named Levy-Ito Model (LIM). The α-stable Levy distribution exhibits a heavy-tail property, enabling the Levy-Ito Model to achieve improved mode estimation, sample diversity, and faster convergence in terms of neural function evaluations (NFEs) compared to SBGMs employing Brownian motion. For the first time, this paper proves the time-reversal formula of stochastic differential equations (SDEs) with Levy processes. It also establishes the sampling equation and presents a numerical solver known as v-Euler-Maruyama for LIM. Furthermore, the paper introduces fractional denoising score matching, which elucidates the training process of LIM. Empirically, the proposed method shows superior mode estimation, diversity in image generation and imputation, and faster convergence of LIM compared to previous SBGMs.

**Strengths:**

In contrast to the previous works, this paper presents a valuable contribution for non-Gaussian distributions where a rigorous proof of the exact time-reversal formula for stochastic differential equations (SDEs) is provided. The provided derivation establishes a solid foundation for the proposed method.

The paper provides clear and comprehensive proofs and demonstrations of the reverse SDE formula, sampling equation, numerical solver (v-Euler-Maruyama), and fractional denoising score matching. These findings effectively illustrate the construction, training, and utilization of the Levy-Ito Model (LIM) in the context of sampling.

Figure 2 visually compares Brownian motion and the Levy process, clearly highlighting the advantageous characteristics of the Levy process, including its notable large jumps.
Several toy examples effectively illustrate the superior capabilities of the proposed method for mode estimation compared to previous Diffusion Models (DMs).

The paper is well-written and easily readable.

**Weaknesses:**

Enhanced diversity is the primary improvement of the Levy-Ito Model over previous Diffusion Models (DMs). Conducting more experiments on mode estimation and sample diversity would further strengthen the proposed method.

**Questions:**

The primary advantages of LIM appear to arise from employing heavy-tailed noise injections, an approach already investigated in prior works. What are the specific benefits of utilizing an isotropic α-stable distribution compared to other potential heavy-tailed distributions?

Training and sampling from the ImageNet dataset with SBGMs are challenging due to its vast diversity. Considering LIM's improvement in generation diversity compared to the previous DMs, I expect that LIM will meaningfully outperform DMs on ImageNet. Although the paper lacks experiments on ImageNet, future research in this area would be intriguing.

Some minor typos:
- line 156: the the Wasserstein-1 -> the Wasserstein-1
- line 158: v-Euler-Maruyamais -> v-Euler-Maruyama is
- It$\bar{o}$ -> It$\hat{o}$


**Limitations:**

---

> ### Author Rebuttal · Authors · 2023-08-09
>
> Thank you for providing valuable and keen insights to enhance the completeness of our paper.
>
> > Question 1
> >
>
> LIM, Heavy-tailed DSM [Deasy et al., 2021], and Denoising Diffusion Gamma Models [Nachmani et al., 2021] all share the advantage of a faster convergence rate for sampling compared to DDPM [Ho et al., 2020]. Additionally, both LIM and Heavy-tailed DSM excel in mode estimation for imbalanced datasets, as mentioned [Deasy et al., 2021]. However, LIM outperforms significantly other models due to 1) its ability to find the exact drift term in the time-reversal formula, 2) precise and stable training, and 3) guaranteed convergence for reverse sampling.
>
> Since all three distributions follow a Lévy process [Dytso et al., 2018][Wang et al., 2012], the drift term for the time-reversal formula can be computed according to [Conforti et al., 2021]. For Isotropic $\alpha$-stable Lévy processes, which follow symmetric and stable distributions, an exact drift term can be computed from the time-reversal formula (Theorem 4.1). Moreover, training can be stable and accurate without the need for integration, as described by (Theorem 4.3). Furthermore, with the knowledge of the exact drift term, a reverse sampling formula ensuring convergence can be derived (Corollary E.1).
>
> In contrast, the drift terms used in Heavy-tailed DSM and Denoising Diffusion Gamma Models appear as an integral, making it difficult to derive an exact form. Even if integration is used during training or approximations are applied to the drift term, accumulating errors lead to inaccurate and unstable training. Moreover, when using an approximation for the drift term to make training practical, there's no assurance that the distance between the distribution $p_{\text{data}}$ of actual data and the distribution $p_{\theta}$ obtained from reverse sampling aligns within a certain range.
>
> In fact, for Heavy-tailed DSM, despite proposing a modified score function corresponding to the Generalized Gaussian distribution and providing a training method, there is no theoretical foundation proving that the modified score function converges to $p_{\text{data}}$ through reverse sampling. Similarly, Denoising Diffusion Gamma Models also fails to propose an exact drift term and present limitations by not providing theoretical evidence for the convergence of reverse sampling using the proposed score function. In fact, the FID performance of these two papers is significantly lower than that of the existing DDPM performance.
>
> We will incorporate the clear distinctions from the previously mentioned existing heavy-tailed noise methods into the paper revision.
>
> > Question 2
> >
>
> Utilizing ImageNet, which has a larger number of classes compared to CIFAR10, to showcase the diversity of LIM through experimental results seems to be a convincing and effective approach. We conducted performance comparisons between Diffusion model [Song et al, 2020] and LIM using the ADM architecture [Dhariwal et al., 2021] on ImageNet 64x64. The results are summarized in the following table:
>
> | Model | FID ($\downarrow)$ | Precision ($\uparrow$) | Recall $(\uparrow)$ |
> | --- | --- | --- | --- |
> | Diffusion model [Song et al, 2020] | 14.23 | 0.6711 | 0.6932 |
> | LIM ($\alpha=1.8$) | $\textbf{12.97}$ | $\textbf{0.6782}$ | $\textbf{0.6937}$ |
>
> The experimental results confirm that LIM performs better than the Diffusion model [Song et al., 2020] in terms of FID, precision, and recall. However, the difference in recall is slight compared to what was expected. This aspect might be due to only using $\alpha=1.8$, and we can expect more improvement of LIM on ImageNet by searching better alpha values
>
> In the revised paper, we will further compare LIM's performance with heavy architectures like NCSN++ [Song et al., 2020] or DiT [Peebles et al., 2021], using the high-resolution dataset ImageNet 256 with varying $\alpha$.
>
> > Question 3
> >
>
> We sincerely appreciate your thorough review of our paper and your detailed feedback on areas that need improvement. All the typos you mentioned have been corrected within our paper, and we will carefully inspect for any other potential errors to enhance the overall quality and completeness of the paper. Thank you once again.
>
> ---
>
> [Conforti et al., 2021] Time reversal of markov processes with jumps under a finite entropy condition (Stochastic Processes and their Applications, 2021)
>
> [Deasy et al., 2021] Heavy-tailed denoising score matching
>
> [Dhariwal et al., 2021] Diffusion Models Beat GANs on Image Synthesis
>
> [Dytso et al., 2018] Analytical properties of generalized Gaussian distributions
>
> [Ho et al., 2020] Denoising Diffusion Probabilistic Models (NeurIPS 2020)
>
> [Nachmani et al., 2021] DENOISING DIFFUSION GAMMA MODELS
>
> [Peebles et al., 2021] Scalable Diffusion Models with Transformers
>
> [Song et al., 2020] Score-Based Generative Modeling through Stochastic Differential Equations
>
> [Wang et al., 2012] Lévy Measure Decompositions for the Beta and Gamma Processes

---

> > ### Comment · Reviewer_Z4Ey · 2023-08-18
> > **Thanks.**
> >
> > We appreciate your response. We keep our score the same.

---

### Official Review · Reviewer_QCLF · 2023-07-05

**Soundness:** 3 good
**Presentation:** 3 good
**Contribution:** 3 good
**Rating:** 7
**Confidence:** 4

**Summary:**

The paper introduces the Levy-Ito Model, a novel score-based generative model (SBGM) that utilizes the isotropic $\alpha$-Levy process as perturbation noise. The authors highlight that their proposed method is the first continuous-time SBGM to incorporate a heavy-tailed process. They aim to leverage the advantages of heavy-tailed processes observed in various cases. For instance, such processes enhance convergence speed in Markov chain Monte Carlo (MCMC) methods like the Langevin algorithm. Additionally, heavy-tailed processes offer faster mode-hopping behavior compared to Gaussians.

First, the authors establish the exact time-reversed stochastic differential equations (SDEs) with Levy process perturbation noise. The authors emphasize that unlike the Wiener process, whose sample paths should be continuous, the sample paths of the Levy process may be discontinuous (jumps) at multiple locations. Consequently, common differential equation techniques may not be applicable. Addressing several technical challenges associated with this SDE formulation, the paper introduces the "fractional score function," replacing the conventional score function in the drift term of the reverse process.

Furthermore, the paper proposes fractional denoising score matching (fractional DSM) to approximate the fractional score in the reverse process. This approach may serve as a more general version of DSM.

Additionally, the authors derive a probability flow formulation for the reverse SDE, similar to conventional SBGMs. This demonstrates that the proposed method can leverage other fast sampling techniques developed for SBGMs, such as advanced integrators.

Finally, the paper presents several experiments to showcase the proposed methods' effectiveness. For instance, mode estimations and sample diversities are evaluated to compare the proposed method to the previous Wiener process-based approaches.

**Strengths:**

Overall I find that the writing is clear, concise, and well-structured, making it easy for readers to follow the arguments and understand the key points. The authors have succeeded in providing a fresh perspective on the topic, shedding new light on the subject matter and offering valuable contributions to the machine learning communities.

**Weaknesses:**

In light of the overall quality of the paper, the experimental parts would benefit from further refinement.

First of all, the discussion on convergence rate can be improved. For example, regarding the results in Figure 7, the convergence rate may be influenced by various aspects, such as network architectures, noise scheduling, or the quality of the trained models. In this aspect, analysis on toy datasets would be more beneficial to provide better evidence.

Secondly, there is room for improvement in the experiments related to Figure 3. It is important to note that FID utilizes classifiers trained on ImageNet datasets. This raises questions about the suitability of FID in analyzing the results for mixtures of Gaussians. I believe a distance metric like MMD may be more appropriate for this purpose.

Lastly, there should be more in-depth discussions about the experiments on image generation benchmark datasets. In Table 1, the transition from Wiener to Levy process in the CIFAR-10 results shows minimal improvement in "recall." However, considering that other values exhibit more significant variations due to network architecture, it becomes necessary to determine the significance of this difference. It is also worth exploring whether the observed differences could be attributed to variations in noise scheduling. Similar patterns also emerge in the results for the CelebA dataset, warranting further analysis and discussion.


**Questions:**

- It needs to be clarified what evidence supports the claim that conventional SBGM exhibits mode collapsing, as mentioned in lines 28-32 of the paper.
- In lines 213-214, it is stated that noise scheduling was performed using the cosine function. However, later in the paper, it is mentioned that VP (presumably referring to a different noise scheduling method) was used. This inconsistency needs to be clarified about the accurate statement.
- Section 5.2 should precede Section 5.1 to explain better the arguments the paper aims to address. By presenting Section 5.2 before Section 5.1, the authors can provide a clearer context and explanation of the differences they intend to discuss. This reordering would enhance the coherence and logical flow of the paper.
- The positioning of the images in Figure 5 and the corresponding columns in the table appears to be reversed, leading to confusion. The inconsistency between the image positions and table columns can create difficulties understanding and interpreting the data.
- There is a mismatch between the order in Tables 2 & 3 and their actual sequence in the document. This inconsistency can cause confusion and make it challenging for readers to locate and reference the correct tables.
- Regarding the differences presented in Tables 2 & 3, it is unclear whether they can be considered statistically significant. 1. Further statistical tests or discussions of the observed variations are necessary to determine the significance of the findings.

**Limitations:**

Please refer to the comments provided in the weaknesses section.

---

> ### Author Rebuttal · Authors · 2023-08-10
>
> > Weaknesses
> >
>
> We sincerely appreciate your detailed suggestions for possible improvements. As you suggested, we will include detailed experimental results on how the convergence rate varies based on 1) architecture and 2) noise scheduling in the next paper revision. While we aimed to investigate the impact of 1) and 2) using the toy dataset, the limited dimensionality of the dataset hindered us from obtaining significant differences. Therefore, due to time and resource limitations, we regret that we could not provide detailed experimental results with real datasets in this rebuttal. However, we would be more than happy to provide supplementary explanations regarding Figure 3.
>
> The FID in Figure 3 is calculated directly from FID formula without utilizing the embeddings of a specific model. Additional measurements for MMD are also conducted and MMD are measured directly from the samples. Below is a table summarizing the values of FID and MMD measured using Diffusion models and LIM:
>
> |  |  Diffusion models | LIM |
> | --- | --- | --- |
> | FID ($\downarrow $) | 8.312 $\pm$ 0.904 | $\textbf{0.663 $\pm$ 0.376}$ |
> | MMD ($\downarrow $)  | 0.026 $\pm$ 0.003 | $\textbf{0.02 $\pm$ 0.002}$ |
>
> It can be observed that the distribution of generated samples $p_{\theta}$ from LIM is more closer to the ground truth distribution $p_{\text{data}}$.  We will incorporate these details during the paper revision period.
>
> > Question 1
> >
>
> <Mode-collapse issue>
>
> Mode-collapse issue has been pointed out that Diffusion models show significant degradation in terms of fidelity and diversity when dealing with imbalanced datasets where the number of samples per class varies [Qin et al., 2023]. Especially, this issue intensifies for tail classes.
>
> > Question 2
> >
>
> We appreciate your feedback on the terminology that might cause confusion in our paper.
>
> In [Song et al., 2020], the VP-SDE formula is defined as:
>
> $dX_t = -\frac{\beta(t)}{2}X_t dt +(\beta(t))^{\frac{1}{2}}dB_t$
>
> and in that paper, $\beta(t)$ is specifically set to a linear function $\beta(t) = (\beta_1-\beta_0)t+\beta_0$. On the other hand, we extended [Song et al., 2020]'s formula to an isotropic $\alpha$-stable Lévy process as:
>
> $dX_t = -\frac{\beta(t)}{\alpha}X_t dt +(\beta(t))^{\frac{1}{\alpha}}dL^{\alpha}_t$
>
> The specific $\beta(t)$ in our paper is derived from the cosine schedule proposed in [Nichol et al., 2021] and tailored to LIM as $\beta(t) = -\alpha\frac{d \log(\cos(\frac{\pi t}{2}))}{dt}$. We will make sure to provide a clearer version in the revised manuscript.
>
> > Question 3
> >
>
> We sincerely appreciate for your thorough review on our paper. Highlighting motivation task for LIM before introducing sample quality, indeed appears far more reasonable. We believe that your editorial efforts provide us guidance toward a better direction in which we are able to make improvements in terms of both the readability and persuasiveness of the paper. In revision period, we will incorporate the feedback you provided and proceed to revise the paper.
>
> > Question 4
> >
>
> As you mentioned, the locations of the image and table in Figure 5 have been leading confusion. Therefore, we have rearranged their locations to make Figure 5 more understandable. Additionally, we will provide additional detailed explanations for Figure 5 to further enhance the quality of the paper. We sincerely appreciate for your valuable insights.
>
> > Question 5
> >
>
> Thank you for your valuable feedback. We will reposition Table 2 and Table 3 as you recommended to improve readability and clarity.
>
> > Question 6
> >
>
> We sincerely appreciate your keen insights and comments on our paper. We have done our best to answer your valuable questions. It seems that there might have been a lack of experiments to demonstrate LIM's superiority in mode estimation aspects for multi-modal datasets, which is especially evident in Tables 2 and 3. Unfortunately, due to limited resource and time, we were unable to incorporate the feedback into the current rebuttal with actual results. Again we greatly apologize for this limitation.
>
> However, in order to validate the hypothesis that LIM can generate high-fidelity and diverse samples for imbalanced datasets, we are committed to conducting further experiments as soon as possible. We will conduct additional experiments and measure additional metrics to supplement the experimental results, and addressing your valuable feedback.
>
> ---
>
> [Chen et al., 2022] Approximation of the invariant measure of stable SDEs by an euler–maruyama scheme (2022)
>
> [Nichol et al., 2021] Improved denoising diffusion probabilistic models, 2021
>
> [Qin et al., 2023] Class-Balancing Diffusion Models (CVPR 2023)
>
> [Song et al., 2020] Score-Based Generative Modeling through Stochastic Differential Equations (ICLR 2020)

---

### Official Review · Reviewer_9rZc · 2023-07-06

**Soundness:** 4 excellent
**Presentation:** 4 excellent
**Contribution:** 4 excellent
**Rating:** 7
**Confidence:** 4

**Summary:**

Prior score-based/diffusion generative models have been defined by Brownian motion. This paper proposes a method of replacing the continuous Gaussian processes with different processes dependent on the characteristic exponent value. The heavy tail property defined by the Levy process allows for higher chance of making larger steps, thus inducing more diverse and faithful sample of the true distribution of the model than its Weiner process counterpart.

The author shows that the model converges faster due to the Levy process having the capability of making large jumps during the forward and reverse process. The resulting model displays promising qualitative and quantitative results in datasets such as CIFAR10 and CelebA and synthetic datasets.

**Strengths:**

- Overall, the paper is well written/organized with clear explanation on the theoretical foundation.
- Experiments are very well thought out and very useful in showing the advantages of utilizing a Levy stable distribution.
- Image metrics show extremely promising results (i.e., FID score).

**Weaknesses:**

- Table 5 in the appendix shows promising FID scores but are heavily dependent on the optimal alpha value. It would be nice if we could get all alpha FID results for Table 1 and not just the optimal alpha value FID score. This would help us understand whether Levy process performs generally better (i.e., when alpha is between 1 and 2) than the baseline diffusion model (i.e., alpha equal 2) or if this seems more like a sensitive hyperparameter for best results.

**Questions:**

- Seen in Figure 3, why are diffusion models bad at faithfully learning the true data distribution? It would be nice if there were more insight on the lackluster performance of diffusion models based on the Weiner process interpretation. Theoretically, it seems like it should perform just as well as the Levy process, could it be due to the number of steps taken by the ODE solver?
- Can the score function learned from the Levy noising process be compatible with other models such as using this learned score function for DDIM inferencing?
-  Why doesn't the heavy tail distribution create instability to the diffusion model path? Line 53-54 states that the variation becomes smaller as it approaches the sample space but wouldn't this also have adverse affect for finer pixel detailing due to the heavy tail? (I have limited knowledge of Levy process, so I may just have misunderstood). (I am willing to raise up my score)

**Limitations:**

Adequately addresses all limitations.

---

> ### Author Rebuttal · Authors · 2023-08-09
>
> > Weakness 1
> >
>
> Thank you for your valuable feedback. We conducted additional experiments regarding $\alpha$-selection for CIFAR10, and CelebA. Below are the FID results based on different values of $\alpha$ for CIFAR10 and CelebA:
>
> | $\alpha$ | CIFAR10 (32x32) | CelebA (64x64) |
> | --- | --- | --- |
> | 1.2 | 5.15 | 2.99 |
> | 1.5 | 2.86 | 1.57 |
> | 1.8 | 2.44 | 2.85 |
> | Diffusion Model [Song et al, 2020] | 2.44  | 3.21 |
>
> The proposed method's performance may differ with different $\alpha$, which means that the best $\alpha$ can differ from data or resolution. Indeed, this seems intuitively natural since images with low resolution less prefer large jumps, while images with high resolution and multiple modalities can benefit exploration from large jumps. Within similar resolution image datasets, there should be a reasonable range of $\alpha$, and here it is found to be between 1.5 and 1.8.
>
> Table 5 uses the DDPM architecture [Ho et al., 2020], whereas above table employs the NCSN++ deep architecture [Song et al., 2020]. Although the overall tendency of declining FID with lower $\alpha$ values remains consistent, it appears that differences in FID itself arises when using different architectures on CIFAR10.
>
> > Question 1
> >
>
> Thank you for your insightful comments.
>
> The Wiener process follows a light-tailed distribution and has a continuous path, resulting in slow convergence rates. Consequently, smaller step sizes are necessary to reach the correct sample, requiring more steps in total. Diffusion models can converge theoretically to the data distribution $p_{\text{data}}$ when a sufficiently large number of steps are taken. However, during reverse sampling, instead of using the score function directly, a score model is used to approximate it. Theoretically, the quality of the score model and NFE determine the quality of the generated samples. However diffusion models use the score model for reverse sampling, it struggles with accurate mode estimation due to limited exploration range of noises compared to LIM during training the score model. You can observe this tendency in Figure 3.
>
> Successful mode estimation can be achieved by training the score models to capture $p(c)$ for each class $c$ of $p_{\text{data}}$. To do this, the (fractional) score function of the perturbed distribution $p_t(\mathbf{x})$ should be learned. The ability of the forward process to explore the sample space effectively determines the score model's capacity for mode estimation as the ratio between the true distribution $p_{\text{data}}(c)$ and the model's predicted distribution $p_{\theta}(c)$ for each mode is proportional to $\frac{p_{\theta}(c)}{p_{\text{data}}(c)} \propto \frac{\int_{\mathbf{x}\in\mathbb{R}^d} p_{\theta}(\mathbf{x},c) d\mathbf{x}}{\int_{\mathbf{x}\in\mathbb{R}^d}p_{\text{data}}(\mathbf{x},c)d\mathbf{x}}$ [Qin et al., 2023].
>
> In a balanced dataset, where  $p_{\theta}(\mathbf{x},c)$ is learned similarly for each $c$, the convergence of $\frac{p_{\theta}(c)}{p_{\text{data}}(c)}$ to 1 is easier to obtain. However, in cases of imbalanced datasets, the exploration range of the noised data due to the properties of Brownian motion limits the frequency of learning for the minor class $c$. In contrast, the heavy-tailed and discontinuous Lévy process has a wider exploration range than Brownian motion, particularly benefiting the accurate learning of $p_{\theta}(\mathbf{x},c)$ for the minor class (Figure 4).
>
> > Question 2
> >
>
> Thank you for the insightful question. We can indeed apply DDIM inference to our model. This is possible because we have derived the probability ODE represented by the fractional score function (Theorem C.1). By applying the Euler method to (Theorem C.1), we arrive at a sampling formula with a structure similar to the fractional score function form of DDIM inference (Corollary E.2). When employing a pre-trained model on the CelebA dataset, the obtained FID scores by LIM-DDIM according to different NFEs are presented in the below table.
>
> | NFE (FID) | 20 | 50 | 100 | 200 | 1000 |
> | --- | --- | --- | --- | --- | --- |
> | LIM-DDIM ($\alpha =1.5$) | 6.73 | 4.80 | $\textbf{3.95}$ | - | - |
> | DDIM | 6.64 | 5.23 | - | 4.78 | 4.88 |
>
> > Question 3
> >
>
> The extended version of the Forward process for the given $\alpha$ in the general VP-SDE framework is given as follows:
>
> $dX_t = -\frac{\beta(t)}{\alpha}X_tdt + (\beta(t))^{\frac{1}{\alpha}}dL^{\alpha}_t$ -(1)
>
> Here, the weak solution $X_t$ of (1) is given as $X_t = a(t)X_0 +\gamma(t)\epsilon$, where $\epsilon \sim \mathcal{S}\alpha\mathcal{S}(1)$.
>
> For this, $a(t)$ is set to the cosine schedule $a(t) = \cos(\frac{\pi}{2}t)$ independent of $\alpha$. According to the VP-SDE structure, $\tilde\gamma_{\alpha}(t)$ becomes $\tilde\gamma_{\alpha}(t) = (1-a^{\alpha}(t))^{\frac{1}{\alpha}}$. Therefore, when $\alpha_1<\alpha_2$, it holds that $\tilde\gamma_{\alpha_1}(t) \le \tilde\gamma_{\alpha_2}(t)$. In other words, LIM has sufficient ability to move from the sample space to the noise space in spite of small noise coefficient $\tilde\gamma_\alpha(t)=(1-a^{\alpha}(t))^{\frac{1}{\alpha}}$ due to large jump noises.
>
> If we fix $\gamma(t)$ for each $\alpha$ and adjust $\tilde a_{\alpha}(t)$ to correspondingly follow the VP-SDE structure in the forward process such that $\tilde a_{\alpha}(t)=(1-\gamma^{\alpha}(t))^{\frac{1}{\alpha}}$, then for $\alpha_1<\alpha_2$, it holds that $\tilde a_{\alpha_1}(t) \le \tilde a_{\alpha_2}(t)$.
>
> Therefore, as $\alpha$ decreases, a gradual degradation effect occurs for the mean corresponding to the forward process $X_t$ with $a(t)X_0$*.* Regardless of whether we keep $a(t)$ fixed or fix $\gamma(t)$ for various $\alpha$ values, LIM can reach the noise space.
>
> ---
>
> [Ho et al., 2020] Denoising Diffusion Probabilistic Models (NeurIPS 2020)
>
> [Qin et al., 2023] Class-Balancing Diffusion Models (CVPR 2023)
>
> [Song et al., 2020] Score-Based Generative Modeling through Stochastic Differential Equations (ICLR 2020)

---

> > ### Comment · Reviewer_9rZc · 2023-08-21
> >
> > The author has satisfied all my questions, I will raise my score.

---

### Official Review · Reviewer_xP7H · 2023-07-19

**Soundness:** 2 fair
**Presentation:** 3 good
**Contribution:** 2 fair
**Rating:** 5
**Confidence:** 3

**Summary:**

This paper presents a new score-based generative model called Lévy-Ito ̄ Model (LIM) that tackles the challenges of slow convergence rate of Number of Function Evaluation (NFE) and mode collapse in diffusion models when applied to imbalanced data. This model leverages isotropic-stable Lévy processes. Initially, the authors derive a precise stochastic differential equation in reverse-time, driven by the Lévy process, and subsequently establish the corresponding fractional denoising score matching technique. The proposed generative model harnesses the advantageous heavy-tailed characteristics of the Lévy process. The experimental findings demonstrate that LIM achieves faster and more diverse sampling, while maintaining exceptional fidelity when compared to existing diffusion models across a range of image datasets.

**Strengths:**

1. This paper investigates a novel optimal non-Gaussian stochastic process(isotropic alpha-stable Lévy processes) for injecting noise, and prove the exact time-reversal formula of SDEs driven by Lévy process.
2. Based on isotropic alpha-stable Lévy processes， this paper proposes a novel score-based diffusion model called Lévy-Ito ̄ Model (LIM).
3. Compared to existing diffusion models, LIM offers faster and more diverse sampling capabilities while maintaining high fidelity across a range of image datasets.

**Weaknesses:**

1. In P2-L31, the authors make claims about the slow convergence rate and mode-collapse issues of previous score-based diffusion models without providing sufficient explanation or evidence to support these claims.
2. In Figure 3, the authors doesn't explain the significance and differences of the two blue clusters in each of the three different subplots. And how is the FID calculated?
3. The related work section provides limited information and lacks a detailed investigation of the recent developments in the last two years.
4. In P5-L156, there are grammatical errors, such as repetitive words and incorrect prepositions. These should be corrected for clarity and readability.
5. Indeed, the experiments seem somewhat limited as they only validate the results on low-resolution datasets such as CIFAR-10 (32x32) and CalebA (64x64).
6. In Appendix C, on P23, Equation 112 is identified as incorrect. The authors should revise the equation or provide the correct version to ensure the accuracy of the paper.

**Questions:**

Please refer to Weaknesses

**Limitations:**

Yes, the authors adequately discussed the limitations of this paper. However, potential negative societal impacts are not discussed.

---

> ### Author Rebuttal · Authors · 2023-08-10
>
> We appreciate your feedback on our paper. We have done our best to answer your keen questions.
>
> > Weakness 1
> >
>
> <Slow convergence>
>
> The reason for the slow convergence of Diffusion models is that the Brownian motion follows a light-tailed distribution and has a continuous path. Various methods have been proposed to reduce the convergence rate while maintaining a fast sampling process and high fidelity [Liu et al. 2022]. Another efforts have also been made to use distillation to reduce the number of steps [Salimans et al., 2022].
>
> <Mode-collapse issue>
>
> It has been pointed out that Diffusion models show significant degradation in terms of fidelity and diversity when dealing with imbalanced datasets where the number of samples per class varies [Qin et al., 2023] with mode collapse issue for tail classes.
>
> > Weakness 2
> >
>
> Two mixtures of Gaussian distribution is the simplest form of an imbalanced dataset. In subplot (b), it can be seen that for the Diffusion model, the generated samples have a ratio of 5.6:1. However for LIM, the generated samples have a ratio of 11.1:1, which is relatively close to the ground truth. This similarity can be observed not only from the ratio aspect but also from FID and MMD. The FID and MMD values for each model are summarized in the table below.
>
> |  | Diffusion models | LIM |
> | --- | --- | --- |
> | FID$ (\downarrow )$ | 8.312 $\pm$ 0.904 | $\textbf{0.663 $\pm$ 0.376}$ |
> | MMD ($\downarrow $) | 0.026 $\pm$ 0.003 | $\textbf{0.02 $\pm$ 0.002}$ |
>
> FID and MMD are calculated directly without using any embeddings for the sets of true samples and generated samples. LIM demonstrates that the distribution $p_{\theta}$ of its generated samples is similar to the ground truth distribution $p_{\text{data}}$. We will incorporate these insights and make the necessary revisions accordingly.
>
> > Weakness 3
> >
>
> Diffusion models have shown advancements in performance through various approaches such as incorporating guidance [Kim et al., 2023][Song et al., 2021], introducing new architectures [Peebles et al., 2023], and proposing novel training methods [Hang et al., 2023]. Additionally, various attempts to reduce convergence rates have been proposed, such as using ODE solvers or [Lu et al., 2022], Fourier neural Operator [Zheng et al., 2022], and distillation [Salimans et al., 2022]. Despite these advancements, diffusion models still face inherent limitations such as slow convergence rates and the mode-collapse issue in imbalanced datasets [Qin et al., 2023].
>
> There are a few papers that have explored the use of noises other than Brownian motion, such as Denoising Diffusion Gamma models [Nachmani et al., 2021] and Heavy-tailed DSM [Deasy et al., 2021]. Both approaches share the common aspect of utilizing DDPM formulas and heavy-tailed distributions. Heavy-tailed DSM employs a Generalized Gaussian distribution and claims to have strengths for imbalanced tasks. Denoising Diffusion Gamma models use a Gamma distribution for noise injection and highlight the advantage of faster convergence rates for sampling. However, they face challenges in terms of performance compared to standard Diffusion models. We will add this information and proceed to modify the related work section accordingly.
>
> > Weakness 4
> >
>
> We appreciate your feedback regarding the grammatical errors present throughout the paper. We will ensure to correct all of them and address the factors that could potentially compromise the readability of the text.
>
> > Weakness 5
> >
>
> To compare the performance on a high-resolution dataset, we chose the DDPM architecture [Ho et al., 2020], and trained LIM and diffusion models [Song et al., 2020] on the CelebA-HQ dataset (256x256). We measured and compared the FID score for each model.
>
> | Model  | FID |
> | --- | --- |
> | Diffusion model [Song et al] (NFE = 1000) | 11.87 |
> | LIM (NFE = 500) | $\textbf{7.76}$ |
>
> It shows that LIM outperforms Diffusion models for CelebA-HQ. However, since this experiment was limited to the DDPM architecture and conducted only on CelebA-HQ, we plan to conduct additional experiments in the next paper revision.
>
> > Weakness 6
> >
>
> Thank you for pointing out the inconsistencies in our proofs. We will correct the error in Equation 112 and carefully revise all other formulas to make sure everything is correct.
>
> ---
>
> [Deasy et al., 2021] Heavy-tailed denoising score matching
>
> [Hang et al., 2023] Efficient Diffusion Training via Min-SNR Weighting Strategy (2023)
>
> [Ho et al., 2020] Denoising Diffusion Probabilistic Models (NeurIPS 2020)
>
> [Kim et al., 2023] Refining Generative Process with Discriminator Guidance in Score-based Diffusion Models (ICML 2023)
>
> [Liu et al. 2022] Pseudo numerical methods for diffusion models on manifolds (2022)
>
> [Lu et al., 2022] DPM-Solver: A Fast ODE Solver for Diffusion Probabilistic Model Sampling in Around 10 Steps (NeurIPS 2022)
>
> [Nachmani et al., 2021] Denoising Diffusion Gamma Models (2021)
>
> [Qin et al., 2023] Class-Balancing Diffusion Models (CVPR 2023)
>
> [Salimans et al., 2022] Progressive distillation for fast sampling of diffusion models (ICLR 2022)
>
> [Song et al., 2020] Score-Based Generative Modeling through Stochastic Differential Equations (ICLR 2020)
>
> [Song et al., 2021] Denoising diffusion implicit models (ICLR 2021)
>
> [Peebles et al., 2023] Scalable Diffusion Models with Transformers (2023)
>
> [*Zheng* et al, 2022] Fast Sampling of Diffusion Models via Operator Learning (NeurIPS 2022)

---

> > ### Comment · Reviewer_xP7H · 2023-08-17
> > **Response to Authors**
> >
> > The authors had addressed my problem, so I raised my initial rating.

---

### Decision · Program_Chairs · 2023-09-21

**Decision:**

Accept (spotlight)

**Comment:**

All reviewers appreciated the paper and the authors successfully answered the questions raised in the reviews